# Do Different Sources of Knowledge and Multiculturalism of Dental and Medical Students Affect the Level of First Aid Education? Do Medical Stereotypes Exist?

**DOI:** 10.3390/ijerph19148260

**Published:** 2022-07-06

**Authors:** Małgorzata Grześkowiak, Marta Iwańska, Adam Pytliński, Alicja Bartkowska-Śniatkowska, Agnieszka D. Gaczkowska

**Affiliations:** 1Department of the Teaching of Anesthesiology and Intensive Therapy, Poznań University of Medical Sciences, Marii Magdaleny St. 14, 61-861 Poznań, Poland; mgrzesko@ump.edu.pl (M.G.); mrumiejowska@ump.edu.pl (M.I.); apytlinski@ump.edu.pl (A.P.); 2Department of Pediatric Anesthesiology and Intensive Therapy, Poznań University of Medical Sciences, Szpitalna St. 27/33, 60-572 Poznań, Poland; asniatko@ump.edu.pl

**Keywords:** first aid, medical stereotypes, dental students, medical students, medical education, defibrillation

## Abstract

Background: The aim of the study was to assess the impact of having various sources of information in the field of first aid on the level of knowledge of dental and medical students, as well as to recognize if medical stereotypes exist in the domain of first aid. Methods: We tested 818 Native-(N) and English (E)—speaking students of medicine (M) and dentistry (D). The questionnaire was constructed in a way that it could detect the issues which created the biggest challenges to the students. It consisted of both theoretical and clinical questions. The intention was to find out whether there were any medical stereotypes. The students were asked to provide the sources of their knowledge to each question, and information about the presence of first aid classes at school. Results: We found medical stereotypes, but only in the questions pertaining to theory: questions concerning defibrillation, opening the airway in infants and the causes of airway obstruction of an unconscious adult. Correlations were found between the sources of knowledge with answers to the questions in each group of students and between the groups. The sources of knowledge in N students came mostly from school, or the students were not able (NA) to indicate the source of knowledge, but E groups gave out of school courses, mass media and their own knowledge (or from the others), as well as NA answers. Interestingly in ED group, among other answers, students also indicated schools as a source of their knowledge. Conclusions: We confirmed that medical stereotypes among dental and medical students exist, and they were not related to multiculturalism or the use of different sources of knowledge.

## 1. Background

The young people who want to study medicine come from different families and environments. They differ in appearance, character, and knowledge. Evert year at The University of Medical Sciences in Poland, Poznań, we teach about 450 Native-(N) students originally from Poland, and 200 English-(E) speaking students. Our students come from different countries from all continents except South America. They present multicultural backgrounds. For years, besides teaching, we had the opportunity to observe the students: their behavior, opportunity to solve simple medical problems and their argumentation. Occasionally we provoked the students to provide explanation concerning the causes of sudden emergency situations, in order to enable them to argue and to present the pathophysiological changes. We observed that even though they could list the problems, they were not able to explain them, they did not understand pathophysiological disorders or they possessed false information and it was not related to language problems. We suppose that the insufficient knowledge of first aid is still present among the students. In specific life-threatening situations the students present certain patterns of thinking and acting that are incorrect. These false patterns are present both in the group of Native-speaking and English-speaking students, and reflect similar problems. We can compare such patterns of inaccurate thinking to a stereotype. A stereotype, according to social psychology, is a belief about a particular category of people [1]. It can be also an expectation about every person of a particular group. A stereotype can be overgeneralized, inaccurate, and resistant to new information [2]. One must also consider the so-called implicit stereotype, which concerns the subconscious of an individual who has no control or awareness. A stereotype is also any thought which is widely adopted about specific types of individuals or certain ways of behaving [3]. What is interesting, is that these thoughts/beliefs may or may not accurately reflect reality [4]. Similarly to the stereotype, one can introduce the concept of a medical stereotype that would function similarly to psychological reductionism (i.e., on the basis of several features of phenomena or persons, a false, reduced conclusion is drawn) [5,6]. In our study we suggest using the term “medical stereotype” which is understood here as false patterns of medical thinking and acting, which do not reflect real procedures, is overgeneralized, inaccurate, and very often resistant to current knowledge. Because we often encounter stereotypes while teaching first aid, we wanted to investigate their causes and find out their sources, so we decided to explore the origin of medical stereotypes in a group of dental and medical students. There is a lack of literature about this topic, with the exception of some authors who presented the stereotypes of first aid in epilepsy and epistaxis [7,8,9]. The others assessed the stereotypes of defibrillation or AED, but they did not examine the knowledge of understanding of the mechanism of defibrillation [10,11]. Similarly, about upper airway obstruction, nobody tested the goal of a maneuver. Our attempt was to investigate in great detail the origin of this problem.

It is worth mentioning here about the main differences in theory/practice between explicit and implicit knowledge. Explicit (conscious) knowledge means that a given person has conscious access to it and can recall it. The strength of recreating this knowledge depends, among other things, on the number of repetitions and the strength of writing (the more often it is played, the better it is encoded). It depends on the neural quality (the age of the person, their condition, fatigue). This knowledge is not a threat to the human ego apparatus. Implicit (unconscious) knowledge—these are information, stimuli that are located in the unconscious sphere, they are difficult to recreate and recall because they may be poorly recorded or threatening, i.e., subject to defense mechanisms, e.g., denial. They are poorly recorded due to the lack of learning, repetition. They leaves a weak memory mark or are emotionally difficult, therefore they are repressed or denied. This knowledge may be distorted. In the aspect of first aid, explicit knowledge plays an important role. The more it is repeated, the more it is recalled. It is worth noting that organizations developing guidelines for life-threatening situations, such as the European Resuscitation Council and the American Heart Association, recommend cyclical repetition of courses, including first aid, every 2–3 years. This obligation also rests with professionals.

Our goal was to assess the impact of various sources of information on the level of knowledge in the field of first aid among dental and medical students. In particular we wanted to direct our focus on dental students and investigate the presence of medical stereotypes among them. Additionally we wanted to determine whether medical stereotypes exist in area of first aid, and if they do, to determine their sources, the influence of origin, residence and multiculturalism of the students.

## 2. Methods

After an approval from the Bioethical Committee (Bioethical Committee by the University of Medical Sciences, Number: 161/18) and the students’ agreement, we tested 818 respondents (NM—native medical 571, ND—native dentistry 58, EM—English medical 125, ED—English dentistry 64 students). The studies were conducted for four years, excluding online classes during the SARS-CoV-2 pandemic. We invited all first year students to participate in the study, and those who did not consent were excluded.

The N students were native Polish citizens, while E students came from 27 different countries (Europe—12 countries, North America—2 countries, West Asia—6 countries, East Asia—4 countries, Africa + Australia—4 countries) who enrolled to study dentistry and medicine here at our university. Their education, culture and environment were different from ours and that provided us with a rich source of data for our study. Based on our observations, we constructed a questionnaire in such a way that it could check mostly these issues with which in our opinion students had the greatest problems.

There were three categories of questions—demographical, theoretical and clinical. The three theoretical questions checked students’ understanding of knowledge. We asked to (1). Give a definition of defibrillation and explain how it works (2). To explain why maximal tilting of the head in infants is forbidden while opening the airway (3). Give a common cause of airway obstruction of unconscious victim lying in supine position. The three clinical questions checked their ability to perform first aid—procedures, maneuvers. We asked them to administer first aid (4). to a choking infant, (5). in seizures, (6). in epistaxis. To avoid any suggestions we used an open ended questionnaire. The answers were coded according to the earlier prepared list of categories. In the first part of the questionnaire, students were asked to name the source of their knowledge to each question, but if they were not able to do it they did not write anything (it was tantamount to the inability to identify the source of knowledge). In the second part we collected information about the existence of first aid classes in school, competency of the teacher and level of teaching as well. Both groups of students were tested at the beginning of the classes in their first year of medical studies.

The following research hypotheses were formulated:The medical stereotypes concerning giving first aid in the group of dental and medical students exist and differ between medical and dental students.The sources of stereotypes in the field of first aid are difficult to identifyThe multiculturalism of dental and medical students has influence on stereotypical answers in the domain of first aid.

For statistical analysis we relied on the program Statistica 12. The following tests were used: U Mann-Whitney, Chi^2Pearson, Fisher-Freeman-Halton, ANOVA, and Kruskal-Wallis. The statistical analysis was completed by a researcher from our university who checked the normality of the distribution and adjusted the appropriate tests. In the case of small numbers (<5), they were not taken into account.

## 3. Results

The demographic data are presented in Table 1. The four groups of students differ, but the most important difference was connected with the presence of first aid classes at school (NM and ND students had the classes, while less than 30% of EM and ED students declared them).

Total knowledge (checked by 6 questions) between NM, ND, EM and ED students was compared using the Kruskal-Wallis test and a correlation was found. This is presented in Figure 1.

We took into consideration answers to each question and compared them among the students. The correlation was found in each question, which is presented in Table 2.

Tests used for statistical analysis: C (Chi^2Pearson), F (Fisher-Freeman-Halton). The statistical correlations found between the groups showed the various levels of correct answers.

We found the correlations between the sources of knowledge with answers to the questions in each group of students (N and E) and between the groups (Table 3). The most often answer in N students was school, or the students were not able (NA) to indicate the source of knowledge, but E groups gave out of school courses, mass media and their own knowledge (or from the others), as well as NA answers. Interestingly in the ED group, among other answers, students also indicated school.

When analyzing the answers of the students, we found medical stereotypes in 3 questions—those that dealt only with the theoretical aspects (Table 4). Interestingly, dental students did not give stereotypical answer explaining defibrillation (ND < 19%, ED < 24%) which is surprising when compared with medical students.

## 4. Discussion

After a very detailed analysis along with of the rest of the data, we found medical stereotypes, but only in the questions testing theory: questions concerning defibrillation, opening the airway in infants and causes of airway obstruction of unconscious adults. A stereotype is a common myth; false knowledge which spreads among populations and nobody knows where the source of a problem is. No literature is available to compare with our results except the problem with defibrillation. In common thinking, myths about a mechanism of defibrillation are that it they provide stimulation to the heart, but in our study, dental students did not indicate stereotypical understanding of defibrillation. It is possible that dental students consider themselves as less resuscitation experts than medical students, and they rely more on proven sources of knowledge than on their beliefs. This false information can be found on the internet, and even on medical pages. The useful expression for the explanation of defibrillation is “restart of the heart” and it is incorrect. Restart means start again. So it can be understood that defibrillation is a stimulation of the stopped heart. Even in medical articles, this term was used [7]. In the Guidelines for CPR and Automated External Defibrillators WebMD available in internet was written that AED is a device “used to restart a heart that has stopped beating”. And even on an official website of the United States government [9] it can be read “AEDs: How These Devices in Public Places Can Restart Hearts” [9]. This is frightening because mass media plays a negative role by spreading false information. The defibrillation procedure is often shown in the movies, but the indications for its implementation are incorrect.To avoid this, in our opinion, a medical consultant should evaluate medical information.

Teachers have a similar and very important role in spreading information. We showed their role observed in native students attending the classes in secondary school. In the opinion of the majority of the students participating in first aid classes at school, the education of a teacher was probably not adequate for such classes and the standard of classes was medium and low. Similar results were presented in the article in which the authors showed that only 21.03% of the 310 high school tested teachers ever participated in life support courses; most of them did not possess adequate knowledge about adult cardiac arrest and foreign body airway obstruction. They found the relationship—as the age of the teachers’ increases the ratio of correct answers decreases. In their opinion teachers are less motivated to be kept updated [10]; the same observations came from Turkey. The majority of the 312 tested primary school teachers did not have enough knowledge about first aid. Those who had knowledge indicated driver’s license courses and printed media as the sources of that knowledge [11]. Getting used to and allowing the functioning of stereotypes leads to shortcuts, and this probably justifies teachers who do not see the need to update their knowledge.

The answers for the clinical questions (first aid in epistaxis, seizures and choking infants) were at different level in the tested groups. Native students (medical – NM and dental -ND) knew what to do in epistaxis (NM 83.54% and ND 89.66%) but E students did not possess adequate knowledge (EM 26.4% and ED 46.88%). We compared these results with a study from Saudi Arabia (SA) [12] where 39.4% of SA students recommended sitting in a position with the head forward, and these results are similar to our E students. The authors concluded that the level of knowledge of the 314 medical students between 18 and 26 years of age was good.

In an article coming from Nigeria [13], authors presented that 50.6% of students wanted to insert an object into the victim’s mouth during seizures which stands in contrast to all studied groups. Dental groups were better than medical groups (ND 0%, ED 1.56%, NM 9.64%, ED 8%). This article was published in 2013, so it may be that time played an important role in these results. The authors from the UK concluded that age of the participants was a significant factor, because in a group >65 years old, 57% thought that putting something into a victim’s mouth would be proper [14]. The older generations (elderly people) spread outdated knowledge. One of the mechanisms responsible for the transmission of stereotypes is the processes of social learning. The child can learn the stereotype directly from the parents when they express their opinions. The child learns them from adults very quickly, uncritically. This mechanism of social learning and modeling is also responsible for the transmission of stereotypical views from generation to generation. Stereotypes are also persistent because, as cognitive schemas, they are very resistant to change [15,16].

Many students participating in our study during the process of filling out the questionnaire did not indicate any source of knowledge. This means that they were not able to do it because they were not aware of where they got this information from (the impact of tacit knowledge). On the other hand, a large group of foreign students did not provide the source of their knowledge. Perhaps the school has too little influence on the transfer of medical knowledge in the field of first aid, since students did not indicate it as the main source of knowledge. From a psychological point of view it should be considered that spreading information among closely related people such as family members, relatives and friends is common without any considerations [2]. That way, old generations transmit outdated knowledge to the next generations. It can be one of the causes of persisting false information.

There are limitations of the study. First of all, tested students did not give answers to all questions, and particularly they were not able to present sources of knowledge. We did not focus on the cultural diversity as much as possible. We did not investigate personal experience, religious, beliefs etc.

## 5. Conclusions

We confirmed that medical stereotypes in the group of dental and medical students exist, and they were not related to multiculturalism or the use of different sources of knowledge. We did not find any obvious sources of knowledge for the medical stereotype except the role of schools, mass media and information from others. We discovered the medical stereotypes surrounding first aid in the theoretical questions concerning: the understanding of a procedure of defibrillation (except dental students), the maneuvers to open the airway in infant and a cause of airway obstruction of unconscious victim. This means that there is still lack of understanding of the pathophysiology of emergency and the mechanisms of recommended maneuvers resulting from the lack of an effective transfer of knowledge. Two things are enough to successfully fight stereotypes: reliable knowledge and critical thinking.

Still, improper phrases are present in medical literature, so based on our research we suggest to replace the term “restart of the heart” with the term “stop of the pathological heat rate”.

Stereotypes simplify the transfer of information and consequently lead to shortcuts. To exclude existing medical stereotypes, we suggest to educate teachers at schools, and to keep control over mass media when they spread medical information or increase the share of professional knowledge sources in providing up-to-date information on first aid. We advise the teachers to attend the review courses on first aid. This is necessary because every five years new guidelines are published.

Nowadays, when surfing the internet, we must proceed with caution while trying to find out medical information. We recommend only professional web sites.

After a detailed analysis of the results, we can verify the research hypotheses.

We confirmed the medical stereotypes in the area of first aid in the theoretical questions concerning: understanding a procedure of defibrillation (except dental students), the maneuvers to open the airway in infant and a cause of airway obstruction of unconscious victim. The stereotypical answers did not differ between medical and dental students except the understanding of defibrillation, which were better in dental students.Our hypothesis was correct and we did not find any obvious sources of knowledge for the medical stereotypes, except the role of schools, mass media and information from the others.We haven’t found any influence of multiculturalism of all tested students coming from 28 countries on stereotypical answers in the domain of first aid. Despite the cultural diversity of the students, this factor did not affect the responses and medical stereotypes were similar among the students.

## Figures and Tables

**Figure 1 ijerph-19-08260-f001:**
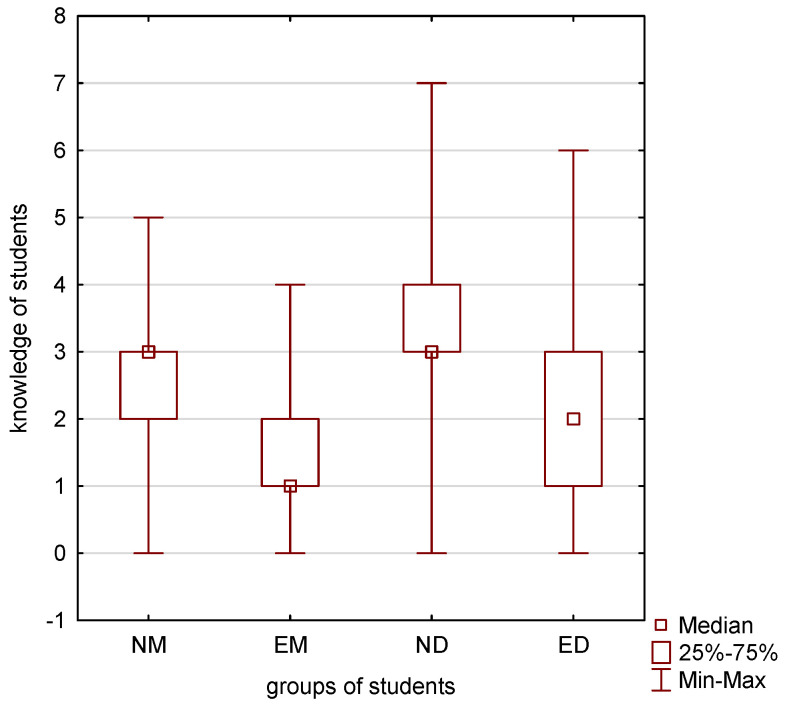
Comparison of total knowledge between NM, ND, EM and ED students (Kruskal—Wallis test, *p* < 0.001).

**Table 1 ijerph-19-08260-t001:** Demographic data and information about first aid classes at school.

	NM Students 571 (100%)	ND Students 58 (100%)	EM Students 125 (100%)	ED Students 64 (100%)
**Gender**				
Female	353 (61.82%)	44 (75.86%)	64 (51.20%)	27 (42.19%)
Male	198 (34.68%)	12 (20.69%)	39 (31.20%)	32 (50.00%)
Not given	20 (3.50%)	2 (3.45%)	22 (17.60%)	5 (7.81%)
**Age**				
18–19	166 (29.08%)	5 (8.62%)	26 (20.80%)	18 (28.12%)
>19–20	236 (41.33%)	17 (29.31%)	8 (6.40%)	14 (21.28%)
>20–21	110 (19.26%)	32 (55.17%)	17 (13.60%)	12 (18.75%)
>22–23	30 (5.25%)	3 (5.17%)	37 (29.60%)	4 (6.25%)
>24–25	9 (1.58%)	1 (1.73%)	18 (14.40%)	9 (14.06%)
>25	6 (1.05%)	0 (0.00%)	15 (12.00%)	6 (9.38%)
Not given	14 (2.45%)	0 (0.00%)	4 (3.20%)	1 (1.56%)
**Place of Residence**				
<10 tsd	150 (26.26%)	19 (32.76%)	7 (5.60%)	3 (4.69%)
10–<100 tsd	170 (29.78%)	16 (27.59%)	30 (24.00%)	7 (10.94%)
100–<500 tsd	104 (18.21%)	13 (22.41%)	10 (8.00%)	12 (18.75%)
>500 tsd	130 (22.77%)	10 (17.21%)	61 (48.80%)	34 (53.12%)
Not given	17 (2.98%)	0 (0.00%)	17 (13.60%)	8 (12.50%)
**First Aid Classes at School**				
Yes	468 (81.97%)	48 (82.76%)	30 (24.00%)	19 (29.69%)
No	99 (17.33%)	10 (17.24%)	93 (74.40%)	42 (65.62%)
Not given	4 (0.70%)	0 (0.00%)	2 (1.60%)	3 (4.69%)

NM—native medical, ND—native dentistry, EM—English medical, ED—English dentistry.

**Table 2 ijerph-19-08260-t002:** Answers to the questions and statistical analysis.

	NM Students 571 (100%)	ND Students 58 (100%)	EM Students 125 (100%)	ED Students 64 (100%)	Statistical Analysis
**Defibrillation means:**					***p* = 0.0000 (C)**
use of a current	421 (73.73%)	52 (89.67%)	67 (53.60%)	45 (70.30%)	
use of AED	79 (13.84%)	1 (1.72%)	7 (5.60%)	0 (0.00%)	
curry out comments from a defibrillator	2 (0.36%)	0 (0.00%)	0 (0.00%)	0 (0.00%)	
other (not correct)	36 (6.30%)	2 (3.44%)	24 (19.20%)	9 (14.07%)	
not given	33 (5.77%)	3 (5.17%)	27 (21.60%)	10 (15.63%)	
**Summary:**					
No. of students who know	502 (87.92%)	53 (91.38%)	74 (59.20%)	45 (70.31%)	
No. of students who do not know	69 (12.08%)	5 (8.62%)	51 (40.80%)	19 (29.69%)	
**Defibrillation is:**					***p* < 0.001 (F)**
depolarisation/reset of the heart	23 (4.03%)	5 (8.62%)	20 (16.00%)	12 (18.75%)	
stimulation of the heart	359 (62.87%)	11 (18.96%)	45 (36.00%)	15 (23.44%)	
action to the sinus node system	27 (4.72%)	0 (0.0%)	8 (6.40%)	1 (1.56%)	
other (not correct)	27 (4.72%)	14 (24.14%)	13 (10.40%)	11 (17.19%)	
not given	135 (23.66%)	28 (48.28%)	39 (31.20%)	25 (39.06%)	
**Summary:**					
No. of students who know	23 (4.03%)	5 (8.62%)	20 (16.00%)	12 (18.75%)	
No. of students who do not know	548 (95.97%)	53 (91.38%)	105 (84.00%)	52 (81.25%)	
**Maximal tilting of the head in infant is forbidden because:**					***p* = 0.0000 (C)**
can close the airway	34 (5.95%)	29 (50.00%)	15 (12.00%)	12 (18.76%)	
can damage the spine/spinal cord	259 (45.37%)	8 (13.79%)	24 (19.20%)	20 (31.25%)	
can cause more damage/infant is more fragile	48 (8.40%)	3 (5.17%)	7 (5.60%)	6 (9.37%)	
can close carotid artery	5 (0.87%)	0 (.00%)	0 (0.00%)	0 (0.00%)	
other (not correct)	84 (14.72%)	3 (5.17%)	44 (35.20%)	17 (26.56%)	
not given	141 (24.69%)	15 (25.87%)	35 (28.00%)	9 (14.06%)	
**Summary:**					
No. of students who know	34 (5.95%)	29 (50.00%)	15 (12.00%)	11 (17.19%)	
No. of students who do not know	537 (94.05%)	29 (50.00%)	110 (88.00%)	53 (82.81%)	
**In infant who is choking a first aid is:**					***p* < 0.001 (F)**
assess severity of choking	1 (0.17%)	11 (18.96%)	0 (0.00%)	6 (9.38%)	
head lower/shake	21 (3.68%)	2 (3.45%)	1 (0.80%)	2 (3.12%)	
tap/blow in the back only	328 (57.45%)	7 (12.07%)	38 (31.20%)	14 (21.87%)	
shake by grabbing the legs	32 (5.60%)	1 (1.72%)	0 (0.00%)	2 (3.12%)	
do the maneuvers immediately	9 (1.57%)	9 (15.52)	8 (6.40%)	13 (20.31%)	
other (not correct)	111 (19.44%)	24 (41.38%)	46 (36,80%)	18 (28.13%)	
not given	69 (12.09%)	4 (6.90%)	31 (24.80%)	9 (14.06%)	
**Summary:**					
No. of students who know	1 (0.17%)	11 (18.96%)	0 (0.00%)	6 (9.37%)	
No. of students who do not know	570 (99.83%)	47 (81.04%)	125 (100.00%)	58 (90.63%)	
**In adult with seizures a first aid is:**					***p* = 0.0000 (C)**
stabilize the head	308 (53.95%)	43 (74.14%)	7 (5.60%)	15 (23.44%)	
put sth in the mouth	55 (9.64%)	0 (0.00%)	10 (8.00%)	1 (1.56%)	
put soft material below the head	47 (8.23%)	10 (17.24%)	5 (4.00%)	3 (4.69%)	
recovery position	37 (6.47%)	0 (0.00%)	18 (14.40%)	4 (6.25%)	
call for help	10 (1.75%)	0 (0.00%)	0 (0.00%)	2 (3.12%)	
other (not correct)	95 (16.64%)	2 (3.45%)	49 (39.20%)	19 (29.69%)	
not given	19 (3.32%)	3 (5.17%)	36 (28.80%)	20 (31.25%)	
**Summary:**					
No. of students who know	308 (53.95%)	38 (65.22%)	5 (4.00%)	15 (23.44%)	
No. of students who do not know	263 (46.05%)	20 (34.48%)	120 (96.00%)	49 (76.56%)	
**A cause of airway obstruction of unconscious adult lying in supine position is:**					***p* < 0.001 (F)**
soft palate	31 (5.43%)	21 (36.22%)	2 (1.60%)	6 (12.50%)	
tongue	340 (59.55%)	22 (37.93%)	44 (35.2%)	21 (32.80%)	
saliva/excretion	27 (4.72%)	1 (1.72%)	15 (12.00%)	3 (4.69%)	
foreign body	16 (2.80%)	1 (1.72%)	1 (0.80%)	3 (4.69%)	
other (not correct)	80 (14.01%)	5 (8.62%)	20 (16.00%)	10 (15.63%)	
not given	77 (13.49%)	8 (13.79%)	43 (34.40%)	19 (29.69%)	
**Summary:**					
No. of students who know	31 (5.43%)	21 (36.21%)	2 (1.60%)	7 (10.94%)	
No. of students who do not know	540 (94.57%)	37 (63.79%)	123 (98.40%)	57 (89.06%)	
**In epistaxis a first aid is:**					***p* = 0.0000 (C)**
sitting position/bend a head forward	465 (81.44%)	52 (89.66%)	19 (15.20%)	19 (29.68%)	
pressure on a nose	12 (2.10%)	0 (0.00%)	14 (11.20%)	11 (17.18%)	
tilt a head back	24 (4.20%)	4 (6.90%)	19 (15.20%)	5 (7.82%)	
cold compress on a forehead/nose	11 (1.93%)	1 (1.72%)	3 (2.40%)	1 (1.56%)	
call for help	0 (0.00%)	0 (0.00%)	0 (0.00%)	2 (3.12%)	
other (not correct)	46 (8.06%)	1 (1.72%)	32 (25.60%’0	11 (17.19%)	
not given	13 (2.27%)	0 (0.00%)	38 (30.40%)	15 (23.44%)	
**Summary:**					
No. of students who know	477 (83.54%)	52 (89.66%)	33 (26.40%)	30 (46.88%)	
No. of students who do not know	94 (16.46%)	6 (10.34%)	92 (73.60%)	34 (53.12%)	

NM—native medical, ND—native dentistry, EM—English medical, ED—English dentistry. The correct answers are underlined.

**Table 3 ijerph-19-08260-t003:** The sources of knowledge to each question and statistical analysis.

	NM Students 571 (100%)	ND Students 58 (100%)	EM Students 125 (100%)	ED Students 64 (100%)
**Defibrillation**				
School Courses (out of school) Professional sources Mass media Own knowledge (or from the others) Different (many sources given) Other Unknown source (not given)	241 (42.20%) 52 (9.10%) 10 (1.76%) 53 (9.28%) 33 (5.78%) 42 (7.36%) 17 (2.98%) 123 (21.54%)	31 (53, 44%) 0 (0.00%) 0 (0.00%) 3 (5.17%) 7 (12.07%) 9 (15.52%) 0 (0. 00%) 8 (13.80%)	14 (11.20%) 15 (12.00%) 2 (1.60%) 14 (11.20%) 6 (4.80%) 2 (1.60%) 12 (9.60%) 60 (48.00%)	12 (18.75%) 4 (6.25%) 12 (18.75%) 4 (6.25%) 8 (12.50%) 2 (3.12%) 0 (0.00%) 22 (34.38%)
Statistical analysis F test—stereotypies and source of knowledge	*p* = 0.08798	*p* = 0.004951	*p* = 0.4113	*p* = 0.3024
Statistical analysis F test—knowledge and source of knowledge	what it means *p* = 0.006125 how it works *p* = 0.07404	what it means *p* = 0.39 how it works *p* = 0.8058	what it means *p* = 0.6604 how it works *p* = 0.385	what it means *p* = 0.08502 how it works *p* = 0.557
**Opening the Airway in Infants**				
School Courses (out of school) Professional sources Mass media Own knowledge (or from the others) Different (many sources given) Other Unknown source	129 (22.58%) 19 (3.31%) 22 (3.94%) 13 (2.26%) 49 (8.57%) 5 (0.86%) 17 (2.97%) 317 (55.51%)	34 (58.62%) 0 (0.00%) 0 (0.00%) 4 (6.90%) 5 (8.62%) 4 (6.90%) 0 (0.00%) 11 (18.96%)	8 (6.4%) 10 (8.00%) 3 (2.40%) 2 (1.60%) 11 (8.80%) 0 (0.00%) 2 (1.60%) 89 (71, 20%)	12 (20.31%) 4 (6.25%) 9 (14.06%) 6 (9.38%) 7 (10.94%) 1 (1.56%) 0 (0.00%) 24 (37.50%)
Statistical analysis F test—stereotypies and source of knowledge	*p* < 0.001	*p* = 0.3122	*p* = 0.0507	*p* = 0.6436
Statistical analysis F test—knowledge and source of knowledge	*p* = 0.0004832	*p* = 0.0001568	*p* = 0.1468	*p* = 0.7149
**The First Aid of Choking Infant**				
School Courses (out of school) Professional sources Mass media Own knowledge (or from the others) Different (many sources given) Other Unknown source	177 (30.99%) 43 (7.54%) 17 (2.98%) 21 (3.68%) 36 (6.30%) 18 (3.16%) 17 (2.97%) 242 (42.38%)	36 (62.07%) 0 (0.00%) 0 (0.00%) 2 (3.45%) 3 (5.17%) 5 (8.62%) 0 (0.00%) 12 (20.69%)	6 (4.80%) 15 (12.00%) 0 (0.00%) 3 (2.40%) 13 (10.40%) 0 (0.00%) 1 (0.80%) 87 (69.60%)	10 (15.63%) 4 (6.25%) 8 (12.50%) 9 (14.06%) 7 (10.94%) 0 (0.00%) 1 (1.56%) 25 (39.06%)
Statistical analysis F test—stereotypies and source of knowledge	*p* = 0.5763	*p* = 0.1712	*p* = 0.184	*p* = 0.5184
Statistical analysis F test—knowledge and source of knowledge	Not done (only 1 person knew)	*p* = 0.7948	Not done	*p* = 0.0001868
**The First Aid of Adult with Seizures**				
School Courses (out of school) Professional sources Mass media Own knowledge (or from the others) Different (many sources given) Other Unknown source	237 (41.50%) 53 (9.29%) 6 (1.05%) 28 (4.90%) 23 (4.03%) 15 (2.63%) 12 (2.10%) 197 (34.50%)	30 (51.72%) 0 (0.00%) 0 (0.00%) 4 (6.90%) 7 (12.07%) 6 (10.94%) 0 (0.00%) 11 (18.97%)	4 (3.20%) 7 (5.60%) 1 (0.80%) 2 (1.60%) 13 (10.40%) 1 (0.80%) 3 (2.40%) 94 (75.20%)	12 (18.75%) 2 (3.12%) 3 (4.69%) 3 (4.69%) 9 (14.06%) 0 (0.00%) 0 (0.00%) 35 (54.69%)
Statistical analysis F test—stereotypies and source of knowledge	*p* < 0.0001	Not done	*p* = 0.08762	*p* = 0.2656
Statistical analysis F test—knowledge and source of knowledge	*p* < 0.0001	*p* = 0.3362	*p* = 0.06422	*p* = 0.008548
**The Cause of Airway Obstruction**				
School Courses (out of school) Professional sources Mass media Own knowledge (or from the others) Different (many sources given) Other Unknown source	190 (33.28%) 35 (6.13%) 18 (3.15%) 18 (3.15%) 32 (5.61%) 15 (2.62%) 13 (2.27%) 250 (43.79%)	38 (65.51%) 0 (0.00%) 0 (0.00%) 0 (0.00%) 4 (6.92%) 5 (8.62%) 0 (0.00%) 11 (18.95%)	1 (0.80%) 11 (8.80%) 3 (2.40%) 2 (1.60%) 7 (5.60%) 0 (0.00%) 2 (1.60%) 99 (79.20%)	11 (17.18%) 2 (3.12%) 3 (4.68%) 2 (3.12%) 7 (10.94%) 0 (0.00%) 1 (1.56%) 38 (59.38%)
Statistical analysis F test—stereotypies and source of knowledge	*p* < 0.0001	*p* = 0.5502	*p* = 0.07276	*p* = 0.002756
Statistical analysis F test—knowledge and source of knowledge	*p* < 0.0001	*p* = 0.7318	Not done (only 2 students knew)	*p* = 0.46
**The First Aid in Epistaxis**				
School Courses (out of school) Professional sources Mass media Own knowledge (or from the others) Different (many sources given) Other Unknown source	224 (39.23%) 41 (7.18%) 11 (1.92%) 10 (1.76%) 53 (9.28%) 18 (3.16%) 15 (2.62%) 199 (34.85%)	39 (67.24%) 0 (0.00%) 0 (0.00%) 6 (10.34%) 2 (3.45%) 4 (6.90%) 0 (0.00%) 7 (12.07%)	4 (3.20%) 7 (5.60%) 1 (0.80%) 1 (0.80%) 10 (8.00%) 0 (0.00%) 1 (0.80%) 101 (80.80%)	8 (12.50%) 2 (3.12%) 3 (4.69%) 9 (14.06%) 5 (7.81%) 1 (1.56%) 0 (0.00%) 36 (56.25%)
Statistical analysis F test—stereotypies and source of knowledge	*p* = 0.4202	*p* = 0.6554	*p* = 0.2568	*p* = 0.07693
Statistical analysis F test—knowledge and source of knowledge	*p* = 0.000005848	*p* = 0.4059	*p* = 0.01254	*p* = 0.6052

NM—native medical, ND—native dentistry, EM—English medical, ED—English dentistry. The students did not indicate the same sources of knowledge to each question. The statistical correlations found between the groups showed that students based on the different sources of knowledge.

**Table 4 ijerph-19-08260-t004:** Confirmed medical stereotypes in the questions (marked by bold). There is given a number of students who answered the questions in this way.

	NM Students 571 (100%)	ND Students 58 (100%)	EM Students 125 (100%)	ED Students 64 (100%)
**Defibrillation is:** **stimulation of the heart**	**359 (62.87%)**	**11 (18.96%)**	**45 (36.00%)**	**15 (23.44%)**
**Maximal tilting of the head in infant is forbidden because: can damage the spine/spinal cord, infant is more fragile**	**307 (53.77%)**	11 (18,96%)	**31 (24.8%)**	**26 (40.62%)**
In infant who is choking a first aid is: head lower/shake, shake by grabbing the legs	53 (9.28)	3 (5.17%)	1 (0.80%)	4 (6.24)
In adult with seizures a first aid is: put sth in the mouth	55 (9.64%)	0 (0.00%)	10 (8.00%)	1 (1.56%)
**A cause of airway obstruction of unconscious adult lying in supine position is: tongue**	**340 (59.55%)**	**22 (37.93%)**	**44 (35.2%)**	**21 (32.80%)**
In epistaxis a first aid is: tilt a head back	24 (4.20%)	4 (6.90%)	19 (15.20%)	5 (7.82%)

NM—native medical, ND—native dentistry, EM—English medical, ED—English dentistry. The confirmed medical stereotypes and number of students who answered stereotypically are marked by bold.

## Data Availability

The data sets used and/or analyzed during the current study are available from the corresponding author on reasonable request.

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
