# Peer review of "Do Different Sources of Knowledge and Multiculturalism of Dental and Medical Students Affect the Level of First Aid Education? Do Medical Stereotypes Exist?"

_ijerph, 2022, doi:10.3390/ijerph19148260_

Round 1

Reviewer 1 Report

Attached ,  you  can  find my  comments !

Reviewer 2 Report

Thank you for this opportunity to provide feedback to this paper on sources of knowledge and multiculturalism in dental and medical students.

Background

Page 1-2: Typically, the background section comprises scholarly evidence of the topic. Some aspects of this section could be moved to the methods section, for instance “The authors of this article are medical and academic teachers…” 

Page 1: Reference/evidence for this “Our students come from difference countries except South America”. 

Page 1: Add a sentence or two about who is classified as ‘Native’? Is this Indigenous communities or Polish born 1st generation? 

Page 1: Please delete the word ‘possessed’ from sentence “They differ in appearance, character and possessed knowledge.”

Page 1: Add capitals and country here “…The University of Medical Sciences in Poland”

Page 2: Was this for all students and what if this relates to several other issues such as language barriers: “We observed that even though they could list the problems, they were not able to explain them, they did not under-stand pathophysiological disorders or they possessed false information.” 

Page 2: Please rephrase this sentence and do you mean all first year students at the University here? “We concluded that there insufficient knowledge of first aid is still present among the students”

Methods

Page 2: Further clarity is needed here please. How were respondents sampled, recruited or surveyed? Online, in person, was it voluntary? The survey ran for 4 consecutive years? How were students randomly selected? 

Page 2: How was multiculturalism of students measured? The aim was to assess various sources of information and level of knowledge in dental and medical students. What is the ‘residence’ and ‘influence of origin’ and how were these measured? Please clarify.

Page 2: Please insert the full name of the ethical committee, including institutional name. 

Page 2-3: Please add further details about statistical analysis. Were any factors moderated or adjusted for confounding? Under what conditions or parameters were used to determine when to use the Chi-square and the Fisher test? i.e. If expected counts were less than 5? 

Results

Tables: Kindly edit all tables so that each row and column is aligned. Maybe it’s the PDF format? It’s currently difficult to interpret as some figures have been double spaced and others in single line. P values should be given at the bottom of the column not in the middle. Instead of the survey/question aligned to the centre, maybe align this to one side.  

Page 3: “A total knowledge between…” What does this ‘total knowledge’ mean? Do you mean the total results from the three theoretical questions? 

Page 9: Typo - should be a full stop not comma in P value “p=0,000005848

Table 4: Please clarify the type of analysis conducted here. Is this a descriptive table between NM / ND / EM / ED? 

Discussion

As a result of the tables that are not aligned, results are difficult to compare/understand. 

Add a paragraph on the limitations of this study please. 

Page 11: Kindly edit this sentence and add referencing “ The old generation spread outdated knowledge”. Maybe: “Age groups over 65 years ….”

Page 11: Please consider rephrasing this sentence “We were astounded to find out that many students…This means that they were” 

Page 11: References are needed for these claims: “From a psychological point of view it should be considered that spreading information among closely related people such as family members, relatives and friends is common without any considerations” 

Page 11: Change ‘web sides’ to ‘web sites’.

ConclusionThank you for this opportunity to provide feedback to this paper on sources of knowledge and multiculturalism in dental and medical students.

Background

Page 1-2: Typically, the background section comprises scholarly evidence of the topic. Some aspects of this section could be moved to the methods section, for instance “The authors of this article are medical and academic teachers…” 

Page 1: Reference/evidence for this “Our students come from difference countries except South America”. 

Page 1: Add a sentence or two about who is classified as ‘Native’? Is this Indigenous communities or Polish born 1st generation? 

Page 1: Please delete the word ‘possessed’ from sentence “They differ in appearance, character and possessed knowledge.”

Page 1: Add capitals and country here “…The University of Medical Sciences in Poland”

Page 2: Was this for all students and what isn’t it possible that this sentence relates to several other issues such as language barriers: “We observed that even though they could list the problems, they were not able to explain them, they did not under-stand pathophysiological disorders or they possessed false information.” 

Page 2: Please rephrase this sentence and do you mean all first year students at the University here? “We concluded that thereinsufficient knowledge of first aid is still present among the students”

Methods

Page 2: Further clarity is needed here please. How were respondents sampled, recruited or surveyed? Online, in person, was it voluntary? The survey ran for 4 consecutive years? How were students randomly selected? 

Page 2: How was multiculturalism of students measured? The aim was to assess various sources of information and level of knowledge in dental and medical students. What is the ‘residence’ and ‘influence of origin’ and how were these measured? Please clarify.

Page 2: Please insert the full name of the ethical committee, including institutional name. 

Page 2-3: Please add further details about statistical analysis. Were any factors moderated or adjusted for confounding? Under what conditions or parameters were used to determine when to use the Chi-square and the Fisher test? i.e. If expected counts were less than 5? 

Results

Tables: Kindly edit all tables so that each row and column is aligned. Maybe it’s the PDF format? It’s currently difficult to interpret as some figures have been double spaced and others in single line. P values should be given at the bottom of the column not in the middle. Instead of the survey/question aligned to the centre, maybe align this to one side.  

Page 3: “A total knowledge between…” What does this ‘total knowledge’ mean? Do you mean the total results from the three theoretical questions? 

Page 9: Typo - should be a full stop not comma in P value “p=0,000005848

Table 4: Please clarify the type of analysis conducted here. Is this a descriptive table between NM / ND / EM / ED? 

Discussion

As a result of the tables that are not aligned, results are difficult to compare/understand. 

Add a paragraph on the limitations of this study please. 

Page 11: Kindly edit this sentence and add referencing “ The old generation spread outdated knowledge”. Maybe along the lines “Age groups over 65 years ….”

Page 11: Please consider rephrasing or delete “We were astounded to find out that many students…This means that they were” 

Page 11: References are needed for these claims: “From a psychological point of view it should be considered that spreading information among closely related people such as family members, relatives and friends is common without any considerations” 

Page 11: Change ‘web sides’ to ‘web sites’.

Conclusion

Page 11: Unsure how the findings were measured in relation to multiculturalism?   

References

Error/missing information for reference number 6 

Page 11: Unsure how the findings were measured in relation to multiculturalism?   

References

Error/missing information for reference number 6 
